# CCL11 Differentially Affects Post-Stroke Brain Injury and Neuroregeneration in Mice Depending on Age

**DOI:** 10.3390/cells9010066

**Published:** 2019-12-26

**Authors:** Simone Lieschke, Bozena Zechmeister, Matteo Haupt, Xuan Zheng, Fengyan Jin, Katharina Hein, Martin S. Weber, Dirk M. Hermann, Mathias Bähr, Ertugrul Kilic, Thorsten R. Doeppner

**Affiliations:** 1Department of Neurology, University Medical Center Goettingen, Robert-Koch-Str. 40, 37075 Goettingen, Germany; simone.lieschke2011@gmx.de (S.L.); bozena.zechmeister@med.uni-goettingen.de (B.Z.); matteo.haupt@stud.uni-goettingen.de (M.H.); xuan.zheng1988@gmail.com (X.Z.); katharina.Hein@krh.eu (K.H.); martin.weber@med.uni-goettingen.de (M.S.W.); mbaehr@gwdg.de (M.B.); 2Institute of Clinical Chemistry, University Medical Center Goettingen, Robert-Koch-Str. 40, 37075 Goettingen, Germany; 3Department of Hematology, Cancer Center, The First Hospital of Jilin University, 2660 Jiefang Rd., Changchun 130021, China; lxcjfy@163.com; 4Institute of Neuropathology, University Medical Center Goettingen, Robert-Koch-Str. 40, 37075 Goettingen, Germany; 5Department of Neurology, University of Duisburg-Essen, Hufelandstr. 55, 45147 Essen, Germany; dirk.Hermann@uk-essen.de; 6Department of Physiology, Istanbul Medipol University, Ekinciler Caddesi No. 19, Beykoz/İstanbul TR 34810, Turkey; kilic44@yahoo.com

**Keywords:** cerebral ischemia, CCL11, eotaxin-1, neuroprotection, inflammation, stroke, neuroregeneration

## Abstract

CCL11 has recently been shown to differentially affect cell survival under various pathological conditions including stroke. Indeed, CCL11 promotes neuroregeneration in neonatal stroke mice. The impact of CCL11 on the adult ischemic brain, however, remains elusive. We therefore studied the effect of ectopic CCL11 on both adolescent (six-week) and adult (six-month) C57BL6 mice exposed to stroke. Intraperitoneal application of CCL11 significantly aggravated acute brain injury in adult mice but not in adolescent mice. Likewise, post-stroke neurological recovery after four weeks was significantly impaired in adult mice whilst CCL11 was present. On the contrary, CCL11 stimulated gliogenesis and neurogenesis in adolescent mice. Flow cytometry analysis of blood and brain samples revealed a modification of inflammation by CCL11 at subacute stages of the disease. In adolescent mice, CCL11 enhances microglial cell, B and T lymphocyte migration towards the brain, whereas only the number of B lymphocytes is increased in the adult brain. Finally, the CCL11 inhibitor SB297006 significantly reversed the aforementioned effects. Our study, for the first time, demonstrates CCL11 to be a key player in mediating secondary cell injury under stroke conditions. Interfering with this pathway, as shown for SB297006, might thus be an interesting approach for future stroke treatment paradigms.

## 1. Introduction

The pathophysiology of cerebral ischemia implies a plethora of inflammatory signaling cascades, all of which allowing for potential therapeutic interventions [1]. Hence, the cytokine CCL11, known to also pass the intact blood-brain barrier (BBB), has recently been recognized as an interesting target under conditions of CNS diseases not limited to cerebral ischemia alone [2]. CCL11, which is also referred to as eotaxin-1, belongs to the CC-chemokine family known for its role in chemoattracting eosinophils [3]. The latter represents the pro-inflammatory properties of CCL11 which have been reported for the peripheral immune system under various pathophysiological conditions [4]. CCL11 is secreted by a vast number of cells, including lymphocytes, monocytes, endothelial cells, fibroblasts, smooth muscle cells, and CNS residing cells [5,6,7,8]. However, the precise characteristics of CCL11 have not been fully understood, yet.

Cytokines, like CCL11, have a differential impact on the CNS exposed to noxious stimuli, ranging from pro-inflammatory and harmful actions towards promoting neurorestorative processes as indicated by the stimulation of endogenous neural progenitor cells (NPCs) within the stem cell niche [3]. As a matter of fact, endogenous neurogenesis persists within the adult mammalian brain, possibly contributing to neurological recovery under defined circumstances, albeit the majority of NPCs suffers from both low survival rates and reduced differentiation rates [9].

CCL11 predominantly acts via the CCR3 receptor, which has the highest affinity for CCL11, and which is expressed on microglia, astrocytes, and NPCs [5,10]. The levels of CCL11 in both the plasma and the cerebrospinal fluid (CSF) physiologically increase during aging [11]. Interestingly, ectopic administration of CCL11 to young mice significantly impairs neurogenesis, as well as learning and memory via interfering with hippocampal activity under non-ischemic conditions [11,12]. The aforementioned impact of CCL11 on the hippocampus also plays a role under pathological conditions such as depressive disorders and Alzheimer’s disease [13]. Likewise, increased levels of CCL11 are found in neurodegenerative diseases, where CSF concentrations of CCL11 correlate with the disease progress [5,14,15]. On the contrary, increased levels of CCL11 in amyotrophic lateral sclerosis (ALS) negatively correlate with disease progression [16]. Under conditions of experimental autoimmune encephalitis in rats, CCL11 modulates the immune response by reducing both demyelination and cumulative microglial activation [17]. At an advanced stage of this disease, however, increased CCL11 levels appear to positively correlate with the progression of multiple sclerosis [5]. As such, these reports demonstrate the heterogenous roles of CCL11 under pathological conditions.

Recent evidence suggests that CCL11 levels are also upregulated after cerebral ischemia in neonatal mice, which results in promoting migration of NPCs in these mice [18]. Whether or not CCL11 is involved in the regulation of ischemic brain injury of the adult mammalian brain is, however, not yet known. As such, we studied the effect of ectopic CCL11 in both adolescent and adult mice exposed to focal cerebral ischemia with an observation period of four weeks, focusing on post-stroke brain injury, inflammation, neuroregeneration, and neurological recovery. Thereafter, therapeutic effects of antagonizing CCL11 were analyzed with regard to post-stroke acute brain injury.

## 2. Materials and Methods

### 2.1. Experimental Groups

Animals were kept under circadian rhythm and had free access to food and water. Housing of animals and experimental procedures were approved by local authorities (33.0-42502-04-16/2291, Nov/24/2016). All experimental procedures were blinded to experimenters and analysts. Animals were randomly allocated to either treatment paradigms. Male C57BL6/J mice (Charles River Laboratories, Sulzfeld, Germany) with either six weeks or six months of age were subjected to transient middle cerebral artery occlusion (MCAO) for 45 min. Mice received intraperitoneal delivery of 10 µg/kg bodyweight CCL11 (R&D Systems, 420-ME/CF, Minneapolis, MN, USA) or 100 µL of phosphate buffered saline (PBS) at the beginning of the reperfusion and daily on the seven consecutive days after reperfusion. Animals were sacrificed on day 1, day 7, or day 28 post-stroke. In the 28-day survival experiments, mice received 50 mg/kg bodyweight bromodeoxyuridine (BrdU) intraperitoneally from day 8 until day 18. Behavioral tests were performed at day 1, day 3, day 7, day 14, day 21 and day 28 after stroke. All animals were pre-trained 2–3 days before induction of the MCAO in order to ensure proper performance for the behavioral tests in question. Readout parameters on day 1 included assessment of brain injury by means of infarct volumetry, neuronal density, and extent of microglial activation. On the contrary, day 7 analyses included signal pathway analyses using Western blot techniques and assessment of post-stroke immune responses using flow cytometry analysis. In day 28 experimental groups, neuroregeneration as well as post-stroke neurological recovery were analyzed. Experimental groups of adolescent and adult animals are shown in Appendix A.

### 2.2. Induction of Focal Cerebral Ischemia

Focal cerebral ischemia was induced by MCAO as described previously by our group [19]. The mice were anesthetized with isoflurane (1.5–2%) in 0.8 L/min O_2_, and buprenorphine (0.1 mg/kg bodyweight) was injected subcutaneously. During the surgery, the animals were put on a feedback-controlled heating plate using a rectal thermometer to ensure a constant body temperature of 37 °C. An incision in the ventral neck region was made to prepare the right common carotid artery (CCA). A silicon-coated monofilament (Doccol, Sharon, MA, USA) was inserted into the CCA and gently moved forward along the internal carotid artery (ICA) until the proximal branch of the middle cerebral artery (MCA) was reached. The monofilament stayed in situ for 45 min. During the surgery, the regional blood flow was recorded with a Laser Doppler flow device (Perimed, Järfälla, Sweden) and only animals with a dropdown of LDF signal to ≤20% were included in this study. After 45 min of cerebral ischemia, the reperfusion was initiated lasting for the observation period in question as stated afore. At the end of the experiment, animals were anesthetized with 5% isoflurane and euthanized by cervical dislocation.

### 2.3. Infarct Volume Analysis and Immunostaining

After 24 h of reperfusion, mice were sacrificed, and brains were cut into 2 mm thick coronal slices. The tissue was stained with 2% of 2,3,5-triphenyltetrazolium chloride (TTC) to measure the infarct lesion size. The TTC solution was prepared with 37 °C phosphate buffer immediately before use, and the brain slices were incubated for 10 min at room temperature. The brain injury was evaluated by software-based infarct volumetry using ImageJ software (National Institutes of Health, USA). Infarct volumes were corrected for cerebral edema. The infarct area was calculated by subtracting the volume of the non-affected ipsilateral hemisphere from total volume of the contralateral hemisphere. Tissue slices for immunohistochemistry were fixed in cryostat matrix and cryostat sections of a 14 µm thickness were made for immunohistochemistry. The neuronal loss was investigated by immunostaining with a rabbit polyclonal anti-NeuN antibody (1:500, Abcam, ab104225, Cambridge, UK). Microglial activation was assessed with a rabbit polyclonal anti-IBA1 (1:500, WAKO, 019-19741, Osaka, Japan) antibody. The secondary antibody used was a donkey anti-rabbit Cy-3 (1:250, JacksonImmuno, 711-165-152, Ely, UK). Quantitative analyses for immunohistochemical stainings were performed, defining four regions of interest (ROIs) within the subventricular zone (SVZ) and the basal ganglia and co-localized to DAPI staining of cell nuclei by cell counting. Stereotactic coordinates for quantitative analysis in the SVZ were 0.14 mm anterior, 2–3 mm ventral and 1–1.25 mm lateral from bregma. Cell count analysis within the basal ganglia was done at 0.14 mm anterior, 2.5–3.25 mm ventral and 1.5–2.25 mm lateral from bregma. The number of proliferating cells (BrdU^+^ cells) was investigated within the BG and the SVZ by manual cell counting. The analysis was performed with a fluorescence microscope (Zeiss, Jena, Germany).

### 2.4. Flow Cytometry Analysis

Cerebral ischemia leads to central but also to systemic inflammation, depicting a distinct temporal pattern with different cell types of the immune system being activated at specific time points [20]. Hence, both peripheral and central immune responses were analyzed on day 7 from ischemic hemispheres and the blood, respectively. After sacrifice, mice were perfused with PBS in order to remove intravasal blood remnants. The hemispheres were digested with collagenase VIII and DNase I, centrifuged and separated using a Percoll gradient. Leukocytes and inflammatory cells were isolated from the intermediate phase. Cell suspension from brain and blood were blocked with Fc-block (CD16/32 FcX rat anti-mouse IgG, BioLegend, 101302, San Diego, CA, USA) to interrupt non-specific binding and afterwards stained for CD45 (CD45 rat anti-mouse IgG- brilliant Violett 510, BD, 563891, Franklin Lakes, NJ, USA), Ly6G (Ly6G rat anti-mouse IgG- FITC, BD, 561105), CD3 (CD3 rat anti-mouse IgG- PE, BD, 555275), CD19 (CD19 rat anti-mouse IgG-APC, BD, 550992), CD11b (CD11b rat anti-mouse IgG-PE- Cy7, eBioscience, 25-0112-81, Waltham, MA, USA), Ly6C (LY6C rat-anti mouse IgG-V450, BD, 560594). For data analysis FlowJo software was used. The gating strategy used is shown in Appendix A.

### 2.5. Western Blot Experiments

Mice were sacrificed after seven days of reperfusion. After decapitation, brains were shock frozen in liquid nitrogen. Ischemic hemispheres were lysed with a homogenisator, using a lysis buffer that contains 1% Triton-X 100, 1 mM EDTA, 131 mM sodium chloride, 1 mM sodium diphosphate, 1 mM sodium fluoride and protease inhibitors for 10 min. Thereafter, the quantification of the protein concentration was photometrically measured (Pierce^TM^ BCA Protein Assay Kit, Thermo Fisher Scientific, Waltham, MA, USA). Lysates were used for sodium dodecyl sulfate polyacrylamide gel electrophoresis (SDS-PAGE) followed by Western blotting. The following primary antibodies were used after membranes were blocked: rabbit polyclonal anti-LC3 (1:100, Abcam, ab128025), rabbit polyclonal anti-mannose receptor (1:500, Abcam, ab64693), rabbit monoclonal anti-s100b (1:2000, Abcam, ab52642), rabbit polyclonal anti-CDK5 (1:1000, SantaCruz, sc-173), rabbit polyclonal anti-p35 (C-19) (1:1000, SantaCruz, sc-820) and mouse monoclonal anti-alpha Tubulin (1:10000, GeneTex, GTX628802). Secondary antibodies were: HRP-conjugated goat polyclonal anti-mouse IgG (1:10000, Abcam, ab97023), HRP-conjugated goat polyclonal anti-rabbit IgG (1:10000, Abcam, ab97051). The membranes were treated with ECL reagent (Cell Signaling Technology, Danvers, MA, USA) and developed with the imaging system ChemiDoc^TM^ XRS+ (Bio Rad, Hercules, CA, USA). Quantitative analysis was performed software based by EvolutionCapt (Vilber Lourmat GmbH, Eberhardzell, Germany).

### 2.6. Behavioral Testing

The neurological recovery was evaluated by behavioral tests which are well-established and already described in detail elsewhere [21]. Briefly, the corner turn test was performed in order to test for motor coordination deficits. Two vertical boards forming an angle of 30° was used, and the mouse was placed into the corner. Healthy mice leave the corner strictly randomly without any side preference once their whiskers make contact with either side of the corner. On the contrary, stroke mice prefer to leave the corner toward the non-impaired body side, i.e., the right body side. For analysis, the laterality index was calculated after 10 measurements per time point.

For the rota rod test, mice were placed on a rotating cylinder with accelerating velocity (rpm). The maximal velocity was achieved after 260 s with a maximal testing time of 300 s. The time until the animal dropped was measured. For each test day, three runs were performed, and means were calculated.

### 2.7. Immunhistochemical Long-Term Studies

For long-term studies, the animals were sacrificed on day 28 after ischemic stroke, and cryostat matrix embedded brain sections were prepared as stated afore. Sections were accordingly stained for assessment of brain damage, inflammation and neuroregeneration. Therefore, the following primary antibodies were used: rabbit polyclonal NeuN (1:500, Abcam, ab104225), rabbit polyclonal IBA1 (1:500, WAKO, 019-19741), mouse monoclonal anti-Doublecortin (DCX) (1:500, Abcam, ab135349), chicken polyclonal anti-GFAP (1:1000, Merck Millipore, AB5541, Darmstadt, Germany), rabbit monoclonal anti-s100b (1:500, Abcam, ab52642), and rat monoclonal anti-BrdU (1:250, Bio Rad, OBT0030G). The secondary antibodies were as follows: donkey anti-rabbit Cy-3 (1:250, JacksonImmuno, 711-165-152), donkey anti-mouse Cy-3 (1:250, JacksonImmuno, 115-165-164), donkey anti-rat Cy-3 (1:250, JacksonImmuno, 712-165-153), goat anti-chicken Alexa Flour488 (1:250, JacksonImmuno, 103-547-008), and donkey anti-rat Alexa488 (1:250, JacksonImmuno, 712-547-003). Quantitative analyses were performed using the very same ROI as mentioned before. Chronic brain injury and inflammation were evaluated by cell counting, and neurorestorative analysis was investigated by counting co-localized cells related to the cell number of interests.

### 2.8. Blocking of CCR3

One milligram per kilogram (1 mg/kg) bodyweight of the CCR3 antagonist SB297006 (Sigma Aldrich, St. Louis, MO, USA) or 50 µL dimethyl sulfoxide (DMSO) was injected intraperitoneally during the beginning of the reperfusion and daily for 3 consecutive days to adolescent and adult male mice alike. Here, the experimental paradigm was chosen in order to ensure sufficiently high biodistribution of the inhibitor within the CNS. Thereafter, animals were sacrificed and neuroprotection as well as inflammation were analyzed as stated afore.

### 2.9. Statistical Analysis

The results of this study were analyzed by GraphPad Prism 7 software (Graph Pad Software Inc., La Jolla, CA, USA). Data are presented as mean value and standard deviation (SD). The comparison of two groups was performed using the Mann–Whitney U-test. The behavioral tests were performed by a two-way analysis of variance (ANOVA) followed by the Sidak’s multiple comparison test. *p*-values of <0.05 were considered statistically significant.

## 3. Results

### 3.1. CCL11 Aggravates Acute Brain Injury in Adult Stroke Mice

Since CCL11 might affect post-stroke brain injury in an age-dependent manner, we first analyzed whether or not ectopic CCL11 affects infarct volume at the acute stage of stroke. Both adolescent and adult mice were subjected to cerebral ischemia followed by a single intraperitoneal injection of CCL11 at the beginning of reperfusion. In adolescent mice, CCL11 did not affect acute brain injury, as depicted by infarct volumetry on day 1 post-stroke (Figure 1A). Likewise, assessment of neuronal density did not reveal any statistical significance between the CCL11 group and control mice (Figure 1B). In line with this, microglial activation was also not altered in in these mice (Figure 1C).

On the contrary, CCL11 significantly enhanced infarct volumes in adult mice (Figure 1D). The latter was supported by the measurement of neuronal densities, which demonstrated a significantly increased loss of neuronal density in the CCL11 group when compared to controls (Figure 1E). Microglial activation that was also analyzed in these groups, however, depicted a surprisingly decreased state of activation in mice treated with CCL11 (Figure 1F).

### 3.2. CCL11 Impairs Neurological Recovery after Cerebral Ischemia in Adult Mice

As the extent of acute brain injury does not necessarily correlate with the extent of neurological impairment, we next analyzed post-stroke motor coordination deficits using two well-defined behavioral tests. During the observation period of four weeks, no statistical difference between the treatment group and the control group was found in adolescent mice (Figure 2A,B), confirming the results on brain injury (Figure 3). On the contrary, treatment of adult mice with CCL11 resulted in a transiently significant impairment on day 1 and 3 in the rota rod test (Figure 2C) and a transiently significant impairment on day 3 in the corner turn test (Figure 2D), reflecting the duration of CCL11 delivery. No difference, however, was observed at the end of the observation period of four weeks.

Although CCL11 aggravated both acute brain injury (Figure 1) and neurological impairment (Figure 2) in adult stroke mice, the latter does not necessarily imply an increased neuronal cell loss in the long run [21]. We therefore measured the neuronal density (Figure 3A,C) defining 5 regions of interest (ROI) in adolescent and adult mice at four weeks post-stroke. The quantitative analysis revealed no significant difference with regard to the neuronal density between the two treatment paradigms in both adolescent and adult mice.

### 3.3. CCL11 Differentially Regulates Post-Ischemic Neuroregeneration in Adolescent and Adult Mice

Since CCL11 promotes neurogenesis in neonatal rodents as stated afore, we next asked the question whether or not the lacking effect of CCL11 on chronic brain injury (Figure 3) might be a consequence of neurorestorative processes induced by CCL11. Indeed, co-staining with the proliferation marker bromodeoxyuridine (BrdU) revealed an increased expression of both the immature neuronal marker doublecortin (DCX) and the mature neuronal marker NeuN in adolescent mice at four weeks after stroke induction (Figure 4A,B). Immunohistochemical analysis in adult mice, however, did not only show no stimulatory effect of CCL11 on post-stroke neurogenesis but rather displayed a significantly decreased level of neurogenesis in these animals (Figure 4 D).

Even though experimental stroke research rather focuses on neurogenesis than on gliogenesis, both of which mutually affect each other [22]. Using co-staining against BrdU and the glial markers GFAP or s100b, respectively, the extent of post-ischemic gliogenesis was, hence, quantified after four weeks (Figure 5). Likewise, post-stroke gliogenesis was significantly increased in adolescent mice treated with CCL11 in comparison to controls, whereas no impact of CCL11 on gliogenesis was observed in adult animals.

### 3.4. CCL11 Modulates the Central and Systemic Immune Response

The pathophysiology of cerebral ischemia contains a complex string of diverse inflammatory signaling cascades, not solely being harmful to the surrounding ischemic tissue. Even though cytokines such as CCL11 are likely to interfere in this process because of their natural biochemical properties, the influence of CCL11 on post-stroke inflammation have not been sufficiently studied, yet. Consequently, both peripheral (blood) and central (brain) immune responses upon treatment with CCL11 were analyzed in the two mice groups, using flow cytometry on day 7 post-stroke (Figure 6 and Figure 7). Cells were labeled against markers for microglia (CD45^intermediate^) and leukocytes (CD45^high^). The latter were further differentiated in T cells (CD45^high^CD3^+^), B cells (CD45^high^Ly6G^–^CD3^–^CD19^+^), neutrophils (CD45^high^Ly6G^+^), and dendritic cells (CD45^high^Ly6G^–^CD3^–^CD11b^+^).

In adolescent mice, CCL11 differentially regulated post-ischemic immune responses in both the blood and the brain. Interestingly, intracerebral subset of microglia, B lymphocytes, and T lymphocytes were significantly increased after CCL11 treatment (Figure 6A–D). Likewise, subset of lymphocyte was also increased in the blood (Figure 6G,H). The subset of peripheral neutrophils was, however, significantly decreased in mice treated with CCL11 (Figure 6F). The regulation of the peripheral post-ischemic immune response by CCL11 did not differ between adolescent and adult mice. Indeed, adult mice treated with CCL11 also displayed reduced subset of neutrophils and increased subset of both B and T cells in the blood (Figure 7F–H). The central immune response was altered by CCL11 only with regard to B cell numbers; B lymphocyte counts were significantly increased after CCL11 treatment (Figure 7D).

### 3.5. CCL11 Does Not Regulate Post-Ischemic Autophagy or Apoptotic Signaling Pathways

As cerebral ischemia involves a plethora of signaling pathways, not only limited to inflammation but also including cell survival pathways, we next evaluated whether or not CCL11 affects autophagy or apoptotic pathways. Western blot analysis on day 7 post-stroke in both adolescent (Appendix A) and adult (Appendix A) mice revealed no significant difference between CCL11-treated mice and controls, albeit protein abundance of cyclin-depended kinase 5 (CDK 5) was increased after CCL11 treatment in adult mice (Appendix A). However, CDK5 in its active form is a heterodimer consisting of CDK5 and p25. The activation of the CDK5 pathway, which is critically involved in cell survival, is a consequence of phosphorylation by p35 [23]. The ratio of p35/p25 was, however, not altered by the administration of CCL11, neither in adolescent (Appendix A) nor in adult (Appendix A) mice. Hence, a regulation of this pathway upon stroke induction appears to be unlikely. Likewise, protein abundance of the autophagy marker LC3, the mannose receptor and s100b (both involved in immune responses) were not affected by CCL11, either (Appendix A).

### 3.6. The CCL11 Antagonist SB297006 Reverses CCL11-Induced Brain Injury in Adult Stroke Mice

After having analyzed the impact of CCL11 on both adolescent and adult mice as well as its underlying way of action, we next set out to study a possible therapeutic effect of CCL11 receptor blockage after stroke. Thus, after antagonizing the chemokine receptor CCR3 in adolescent stroke mice, acute brain injury was not affected by SB297006 (Figure 8A–C). As expected, neither infarct volume, nor neuronal density, nor microglial activation were regulated under these conditions.

On the contrary, treatment of adult mice with SB297006 did not only reverse the CCL11-induced aggravation of brain injury but also yielded neuroprotection when compared to control mice treated with solvent only (Figure 8D–F). En detail, infarct volumes were decreased, and neuronal densities were increased after CCR3 receptor blockage. Post-stroke microglial activation was, consistent to our findings, increased in adult mice treated with SB297006.

## 4. Discussion

Post-stroke inflammation does not only imply the activation of both immune competent cells and non-immune cells, but also the secretion of cytokines serving as effector molecules in the process. Although the precise role of the cytokine CCL11 still remains unknown, its significant contribution to cell survival signaling pathways is of no doubt [24,25]. Hence, the present work for the first time analyzed the impact of ectopic CCL11 in a model of cerebral ischemia in both adolescent and adult mice. Even though using elderly animals instead of adult mice might have better reflected the clinical situation, the fundamental relevance of the present work is not hampered. On the contrary, the ectopic application of CCL11 in elderly mice might have produced confounding results, since CCL11 itself negatively regulates neurogenesis in elderly mice under physiological conditions [11]. Consequently, ectopic application of CCL11 in elderly mice might even result in higher mortality of such stroke mice.

The work presented herein indicates that CCL11 differentially affects post-stroke brain injury and neuroregeneration, depending on the age of mice exposed to stroke. As such, CCL11 significantly aggravates brain injury in adult stroke mice, without affecting infarct volumes in adolescent mice. On the contrary, neuroregeneration is significantly enhanced in adolescent mice treated with CCL11, as shown by increased levels of both gliogenesis and neurogenesis. The therapeutic potential of the CCL11 pathway under ischemic conditions is underscored by using the CCL11 inhibitor SB297006, which not only reverses the CCL11-induced aggravation of brain injury but also yields neuroprotection in adult stroke mice. Whereas CCL11 is known to cross the BBB [2], information regarding SB297006 on this matter does not exist. To the best of our knowledge, SB297006 has not been used in any in vivo model. Even though we cannot completely exclude a possible systemic effect of SB297006, it stands to reason that the results observed are a consequence of central effects by the inhibitor itself. In this context, one also has to stress that SB297006 was given on three consecutive days following stroke induction, i.e., at a time point were the BBB is open due to the stroke itself. Of note, SB297006 has only been used in a single therapeutic approach, i.e., a combined approach together with CCL11 has not been performed, which would be of interest for further studies.

Pro-inflammatory effects of cytokines like CCL11 in the post-ischemic tissue lead to acute brain injury through their ability to recruit and activate inflammatory cells [3]. Herein, the administration of ectopic CCL11 in adolescent mice had no effect on post-stroke brain injury and on microglia. In adult mice, however, CCL11 significantly enhanced brain injury and reduced microglial numbers. The importance of the CCL11 signaling pathway under such conditions is underlined using the SB297006 antagonist, which reduces brain injury and increases microglial activity. With cerebral ischemia usually known to activate microglia resulting in a state of inflammation [26], decreased numbers of microglia in adult stroke animals after treatment with CCL11 are surprising. This is in contrast to previous reports, which have shown promoted migration and activation of microglia due to CCL11, all of which culminating in high levels of oxidative stress and ultimately leading to cell death [27]. In vitro work by Parajuli et al. using a co-culture system of microglia and neurons identified microglial cells to be a crucial mediator of CCL11-induced cell death [27]. These data appear to be contradictory to the present data set at first glance, where cell injury was pronounced in adult stroke mice by CCL11 in spite of decreased numbers of microglia. Even though an exact explanation regarding this phenomenon might fail, it stands to reason that the role of post-stroke microglia is just about to be understood [28,29]. As a matter of fact, microglial contribution to brain injury after stroke is subtle and by far not always detrimental [30,31]. The latter, of course, is not only restricted to the sheer number of microglia residing in the ischemic brain, but also depends on the state of activation of microglial cells, which implies the differentiation between M1 and M2 phenotypes [26]. Of note, the polarization of microglia and thus the development of different phenotypes is a dynamic process being a consequence of various activation pathways with intermediate states known as well. Herein, microglia were only labeled against IBA1, not necessarily allowing for a differentiation between different states of activation of these cells. Of note, IBA1 does not only stain resident microglia but also all myeloid-lineage cells including CNS border-associated macrophages as well as blood-borne monocytes/macrophages. Consequently, we could not completely distinguish between these different cell types [32]. This restriction in histology impedes the assessment of microglia-driven inflammation. Since the present work focused on performing a proof-of-concept study on CCL11 in adult stroke mice and thus indirectly implied a therapeutic hypothesis, further elucidating the precise role of microglia under such conditions was beyond the scope the present work.

In light of clinical and potential therapeutic implications, studying acute brain injury only is insufficient. Consequently, neuronal densities as well as neurological function were studied for as long as four weeks after stroke induction. Taken into account CCL11 biodistribution patterns and previous CCL11 delivery protocols, long-term experiments were thus performed using a seven-day CCL11 delivery paradigm [11,18,33]. Likewise, CCL11 had no effect on adolescent mice under these conditions. The adult mice treated with CCL11, however, developed a transient impairment of neurological recovery when compared to controls, reflecting the duration of CCL11 delivery. At the end of CCL11 treatment at seven days post-stroke, the behavioral test performance approximated test scores from control mice. Even though behavioral recovery and reversal of acute brain injury of adult mice treated with CCL11 at this late stage of the disease most likely only reflects the seven-day CCL11 delivery protocol, a modulation of endogenous neuro-restorative mechanisms cannot be excluded. Indeed, CCL11 promotes endogenous neurogenesis in neonatal mice [18], whereas CCL11 decreases neurogenesis in elderly mice under otherwise physiological conditions, resulting in an impairment of both learning and memory [11]. Interestingly, both gliogenesis and neurogenesis were significantly increased in adolescent mice after CCL11 treatment, which is in line with previous work [18,34]. Adult mice did not only show no stimulation of neurogenesis after CCL11 treatment, they rather displayed a decreased level of neurogenesis four weeks after stroke. Indeed, microglial cells have been reported to differentially regulate neural progenitor cells within stem cell niches, albeit with different results depending on the localization in the CNS compartment [31,35]. Reports on the regulation of endogenous neurogenesis by microglia under physiological and pathological conditions are, however, still a matter of debate [30,36,37,38,39,40]. Although microglial activation in both adolescent and adult mice treated with CCL11 was not changed at four weeks post-stroke, it stands to reason that post-stroke regulation of gliogenesis and neurogenesis is likely to be a consequence of microglial activity. Of note, CCL11 delivery lasted until week one post-stroke, most likely affecting the generation and differentiation of new-born cells. The latter persist in the case of adolescent mice, albeit CCL11 delivery has already been ceased. In this context, we cannot exclude the possibility of CCL11 having long-lasting effects on post-stroke neuroregeneration beyond day 28. However, such effects are very unlikely when taking into account a high cell death rate of new-born cells after induction of stroke. As a matter of fact, the majority of such cells dies within three months post-stroke due to secondary cell death [21].

Systemically increased CCL11 levels alter the peripheral immune system [4,41], but might also affect the CNS beyond the role of microglia. As such, B cells are known to secrete CCL11 and can in turn be activated themselves by CCL11, resulting in an enhanced Th2 immune response [42,43,44]. Confirming these observations, the subset of B cells in the blood and brain were increased in both adolescent and adult mice. Similarly, CCL11 is reported to activate T cells, which is in line with our data, demonstrating an increased subset of T cells in both animals group in blood [43]. Likewise, the subset of T cells in the brain is also significantly increased upon CCL11 treatment in adolescent mice exposed to stroke. Beyond the analysis of microglia and lymphocytes, there was no significant difference in the subset of neutrophils in the brain due to CCL11 treatment. As a matter of fact, transmigration of immune cells into the CNS compartment requires, however, the activation of endothelial cells. The activation of the latter by cytokines depends, among other, on the interleukin-8/CXCL8 signaling pathway and allows the transmigration of neutrophils into inflammatory tissue [45]. CCL11 has been shown to inhibit this interleukin-8/CXCL8 signaling pathway, causing thereby an altered endothelial cell activation and modified recruitment of granulocytes, which ends up in a reduced migration of neutrophils [45]. In the blood, the increased T and B cells subset led to a proportional decrease of neutrophils due to our gating strategy (Appendix A). As stated afore, a mutual interaction between peripheral and central immune responses under stroke conditions initiated by systemic CCL11 cannot be completely excluded. Even though data on this matter appears to be not present in the literature, a precise analysis of such an interaction was beyond the scope of the present work.

CCL11 exerts a distinct modulation of the immune system that goes beyond the mere involvement of individual immune-competent cells. Consequently, we further studied signaling pathways that might be affected by ectopic CCL11 within the ischemic brain tissue. Microglial signaling involving the mannose receptor, which is constitutively expressed on M2 microglia contributing to cell migration, intracellular signaling, phagocytosis, and remodeling of inflammatory tissue [46], was not altered by CCL11 in the present work. This is in contrast to previous reports on rats exposed to stroke, where the activation of the mannose receptor yielded distinct restorative effects [46]. Pro-inflammatory M1 microglial cell activation can also be achieved via intracellular, calcium-dependent signaling using s100b. Increased levels of the latter were observed within the first two weeks post-stroke in adult mice which were subjected to MCAO, leading to an aggravation of acute brain injury [47]. Herein, however, CCL11 did not affect expression patterns of s100b, suggesting that the s100b signaling pathway is of minor relevance in the present experimental paradigm. Secondary cell injury upon stroke induction involves a plethora of pro-apoptotic signaling pathways [48], among which the activation of the CDK5/p35 pathway is detrimental [23,49]. Nevertheless, the CDK5/p35 pathway was not affected by CCL11, neither in adolescent nor in adult stroke animals. Recent reports also suggest a possible role for autophagy, being presumably regulated under diverse pathological conditions including stroke. Yet, the precise role of stroke-associated autophagy is elusive with autophagy being a double-edged sword that can be either protective or harmful [50]. Autophagy, however, appears to be of no concern in CCL11-aggravated ischemic brain injury of adult mice. Hence, the latter rather is a consequence of CCL11-dependent immune cell regulation than due to modulation of apoptotic or autophagic signaling pathways.

## 5. Conclusions

The present study, for the first time, demonstrates the controversial effects of CCL11 in both adolescent and adult stroke mice. As such, we herein demonstrated CCL11 to promote post-stroke neurogenesis and gliogenesis in adolescent mice, whereas elderly mice displayed aggravated acute brain injury, impaired functional recovery and inhibited neurogenesis. The present study thus highlights the importance of age in a possible therapeutic setting against stroke, when interfering with the CCL11 signaling pathway. Further preclinical studies on elucidating the latter, as well as an analysis of age-dependent CCR3 expression patterns are, however, in order before more translational experimental paradigms or even clinical trials are reasonable.

## Figures and Tables

**Figure 1 cells-09-00066-f001:**
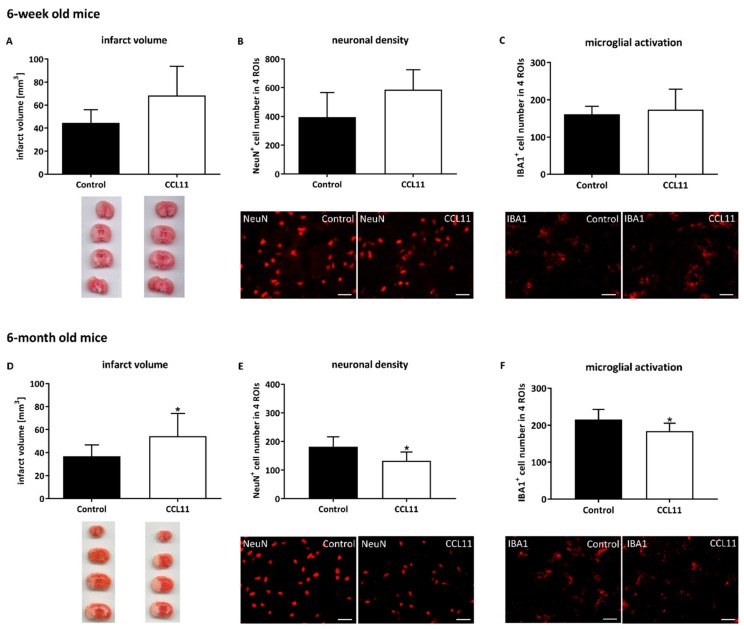
CCL11 aggravates acute ischemic brain injury in adult mice. C57BL6 male mice were subjected to cerebral ischemia for 45 min followed by reperfusion for 24 h. At the beginning of the reperfusion, mice received intraperitoneal injections of either 10 µg/kg bodyweight of CCL11 or 100 µL of PBS as control. (**A**) Infarct volume measurement using TTC staining, ns *p* = 0.059 of the adolescent mice (**B**) assessment of neuronal density as indicated by NeuN^+^ cells, ns *p* = 0.053 of adolescent mice and (**C**) analysis of microglial activation quantifying the number of IBA1^+^ cells, ns *p* = 0.384, of adolescent mice within the ischemic striatum. (**D**) Infarct volume measurement using TTC staining, *significantly different from the corresponding control, *p* = 0.037 of adult mice. (**E**) Assessment of neuronal density as indicated by NeuN^+^ cells, *significantly different from the corresponding control, *p* = 0.036 of adult mice and (**F**) analysis of microglial activation quantifying the number of IBA1^+^ cells, *significantly different from the corresponding control, *p* = 0.030 of adult mice within the ischemic striatum. Photographs depicted below each diagram show representative stainings for each experimental condition. *n* = 10 for each group of the adolescent mice, *n* = 8 in the adult control group, and *n* = 7 in the CCL11 adult group. All data are given as means ± S.D., scale bars 10 µm.

**Figure 2 cells-09-00066-f002:**
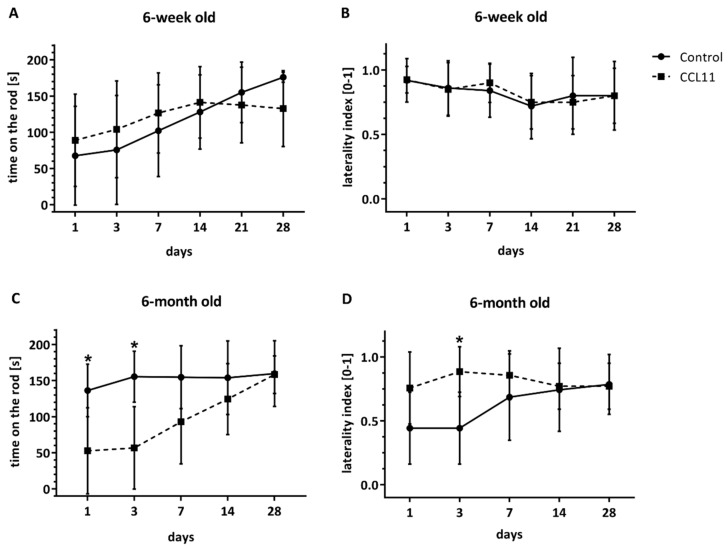
CCL11 impairs post-stroke neurological recovery in adult mice. C57BL6 male mice were subjected to cerebral ischemia for 45 min followed by reperfusion for 28 d. At the beginning of the reperfusion, mice received intraperitoneal injections of either 10 µg/kg bodyweight of CCL11 or 100 µL of PBS as control followed by daily injections on the seven consecutive days after reperfusion. At the time points given, behavioral tests were performed. For assessment of motor coordination deficits, the rota rod test in (**A**) of adolescent mice (ns *p* = 0.965 at day 1, ns *p* = 0.872 at day 3, ns *p* = 0.933 at day 7, ns *p* = 0.997 at day 14, ns *p* = 0.987 at day 21 and ns *p* = 0.496 at day 28) in (**C**) of adult animals (*significantly different from the corresponding control, *p* = 0.008 at day 1, *significantly different from the corresponding control *p* = 0.001 at day 3, ns *p* = 0.085 at day 7, ns *p* = 0.997 at day 14, *p* = 0.758 and ns *p* = 0.999 at day 28) and the corner turn test in (**B**) of adolescent mice (ns *p* = 0.999 at day 1, ns *p* = 0.999 at day 3, ns *p* = 0.993 at day 7, ns *p* = 0.999 at day 14, ns *p* = 0.997 at day 21 and ns *p* = 0.999 at day 28) and (**D**) of adult mice (ns *p* = 0.119 at day 1, *significantly different from the corresponding control *p* = 0.009 at day 3, ns *p* = 0.701 at day 7, ns *p* = 0.999 at day 14, *p* = 0.999 and ns *p* = 0.999 at day 28) were performed. *n* = 8 in the CCL11 group and *n* = 10 in the control group of adolescent mice. *n* = 6 in the CCL11 group and *n* = 7 in the control group of adult mice. All data are given as means ± S.D.

**Figure 3 cells-09-00066-f003:**
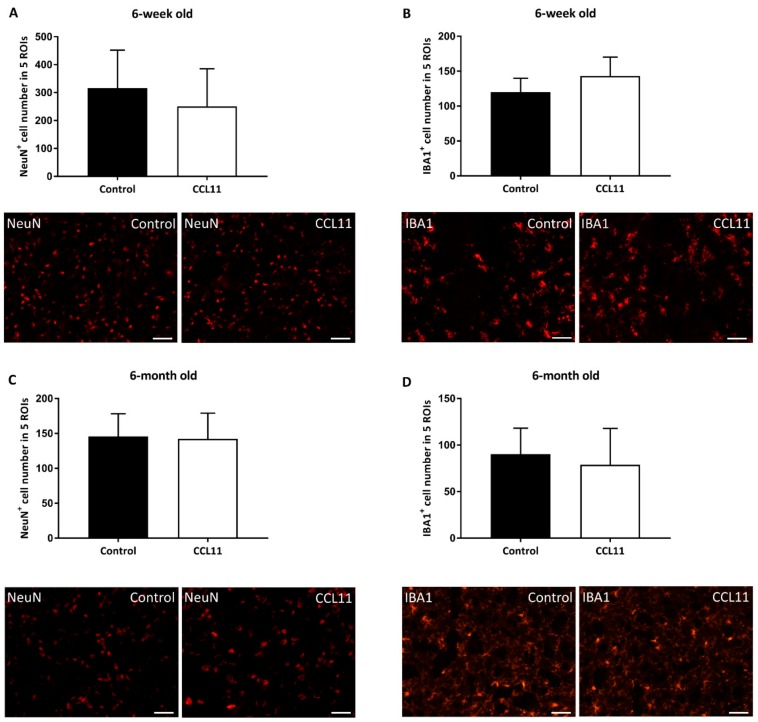
CCL11 does not affect chronic ischemic brain injury. C57BL6 male mice were subjected to cerebral ischemia for 45 min followed by reperfusion for 28 d. At the beginning of the reperfusion, mice received intraperitoneal injections of either 10 µg/kg bodyweight of CCL11 or 100 µL of PBS as control followed by daily injections on the seven consecutive days after reperfusion. Neuronal loss was measured by NeuN^+^ cells in five ROIs in (**A**) adolescent mice, ns *p* = 0.408 and (**C**) adult mice, ns *p* = 0.999. Microglial activation measured by IBA1^+^ cells in five ROIs was done in (**B**) adolescent mice, ns *p* = 0.070 and (**D**) adult mice, ns *p* = 0.655. Representative photographs of stainings are shown below each diagram for each experimental condition. *n* = 8 in the CCL11 group and *n* = 10 in the control group of adolescent mice. *n* = 6 in the CCL11 group and *n* = 7 in the control group of adult mice. All data are given as means ± S.D., scale bars 20 µm.

**Figure 4 cells-09-00066-f004:**
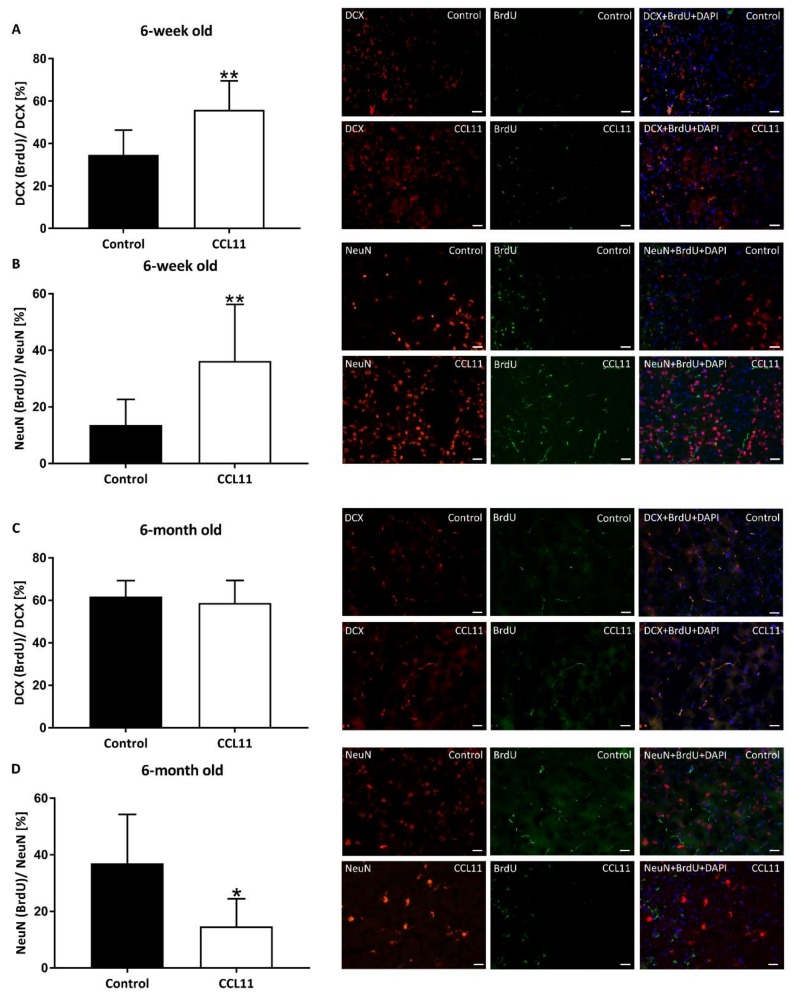
CCL11 differentially regulates post-stroke neurogenesis in adolescent and adult mice. C57BL6 male mice were subjected to cerebral ischemia for 45 min followed by reperfusion for 28 d. At the beginning of the reperfusion, mice received intraperitoneal injections of either 10 µg/kg bodyweight of CCL11 or 100 µL of PBS as control followed by daily injections on the seven consecutive days after reperfusion. From day 8 to day 18, BrdU was intraperitoneally injected in mice. Neurogenesis was evaluated by co-expression analysis of the neuronal markers together with the proliferation marker BrdU in 5 ROIs of both adolescent (**A**,**B**) and adult (**C**,**D**) mice. In (A) DCX and BrdU co-localized cells, **significantly different from the corresponding control, *p* = 0.006, in (**B**) NeuN and BrdU co-localized cells, **significantly different from the corresponding control, *p* = 0.008, in (**C**) DCX and BrdU co-localized cells, ns *p* = 0.445, in (**D**) NeuN and BrdU co-localized cells, *significantly different from the corresponding control, *p* < 0.030. Photographs depicted beside each diagram show representative stainings for each experimental condition. *n* = 8 in the CCL11 group and *n* = 10 in the control group of adolescent mice. *n* = 6 in the CCL11 group and *n* = 7 in the control group of adult mice. All data are given as means ± S.D., scale bars 20 µm.

**Figure 5 cells-09-00066-f005:**
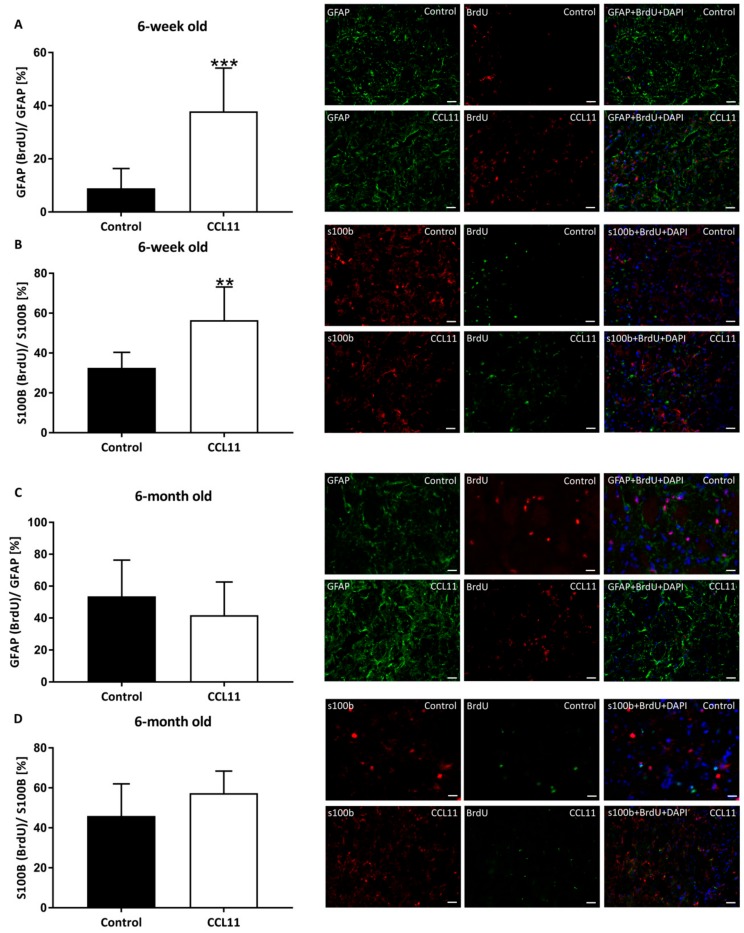
CCL11 promotes post-stroke gliogenesis in adolescent mice. C57BL6 male mice were subjected to cerebral ischemia for 45 min followed by reperfusion for 28 d. At the beginning of the reperfusion, mice received intraperitoneal injections of either 10 µg/kg bodyweight of CCL11 or 100 µL of PBS as control followed by daily injections on the seven consecutive days after reperfusion. BrdU was injected from day 8 to day 18. Gliogenesis was evaluated by co-expression analysis of the glial markers together with BrdU as counted in 5 ROIs in both adolescent (**A**,**B**) and adult (**C**,**D**) mice. Figure (**A**) shows GFAP and BrdU co-localized cells, ***significantly different from the corresponding control, *p* = 0.0005. Figure (**B**) shows s100b and BrdU co-localized cells, **significantly different from the corresponding control, *p* = 0.004. In Figure (**C**) GFAP and BrdU co-localized cells are depicted, ns *p* = 0.329. Figure (**D**) displays s100b and BrdU co-localized cells, ns *p* = 0.137. Photographs depicted beside each diagram show representative stainings for each experimental condition. *n* = 8 in the CCL11 group and *n* = 10 in the control group of adolescent mice. *n* = 6 in the CCL11 group and *n* = 7 in the control group of adult mice. All data are given as means ± S.D., scale bars 20 µm.

**Figure 6 cells-09-00066-f006:**
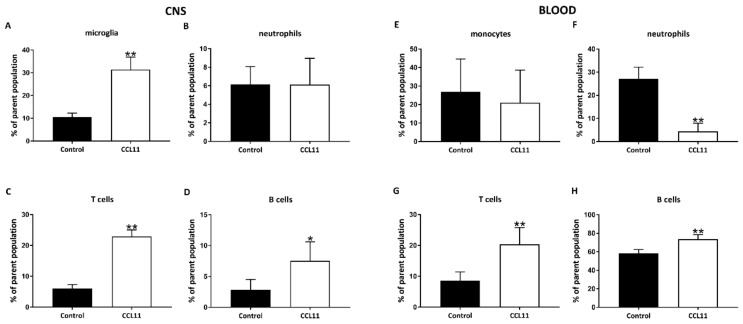
CCL11 modifies subacute central and systemic immune responses in post-stroke adolescent mice. C57BL6 male mice with an age of six weeks were subjected to cerebral ischemia for 45 min followed by reperfusion for 7 days. At the beginning of the reperfusion, mice received intraperitoneal injections of either 10 µg/kg bodyweight of CCL11 or 100 µL of PBS as control and daily on the seven consecutive days after reperfusion. Flow cytometry was analyzed with FlowJo software. (**A**–**D**) shows the quantitative analysis of adolescent animals of (**A**) CD45^int^ cells, **significantly different from the corresponding control, *p* = 0.007. (**B**) shows Ly6G^+^ cells, ns *p* = 0.690. (**C**) displays CD3^+^ cells, **significantly different from the corresponding control, *p* = 0.007, and (**D**) shows CD19^+^ cells, *significantly different from the corresponding control, *p* = 0.031, as measured in the CNS. In (**E**–**H**) the quantitative analysis of the adolescent animals of (**E**) CD45^int^ cells was performed, ns *p* > 0.999, (**F**) depicts Ly6G^+^ cells, **significantly different from the corresponding control, *p* = 0.007. (**G**) shows CD3^+^ cells, **significantly different from the corresponding control, *p* = 0.007, and (**H**) demonstrates CD19^+^ cells, **significantly different from the corresponding control, *p* = 0.007, as measured in the blood. The number of mice was *n* = 5 in each group. All data are given as means ± S.D.

**Figure 7 cells-09-00066-f007:**
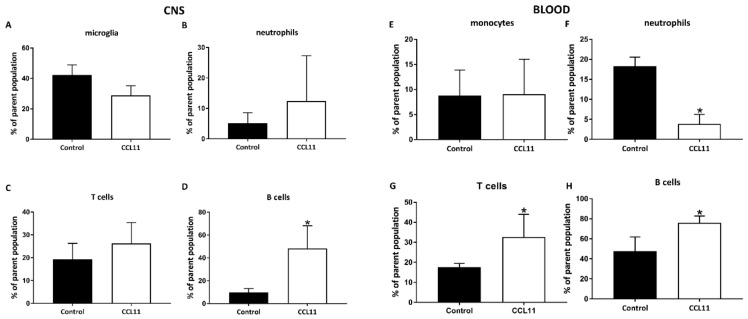
CCL11 modifies the subacute central and systemic immune response in post-stroke adult mice. C57BL6 male mice with an age of six month were subjected to cerebral ischemia for 45 min followed by reperfusion for 7 days. At the beginning of the reperfusion, mice received intraperitoneal injections of either 10 µg/kg bodyweight of CCL11 or 100 µL of PBS as control and daily on the seven consecutive days after reperfusion. Flow cytometry was analyzed with FlowJo software. (**A**–**D**) shows the quantitative analysis of adult animals of (**A**) CD45^int^ cells, ns *p* = 0.057, (**B**) shows Ly6G^+^cells, ns *p* = 0.885, (**C**) displays CD3^+^ cells, ns *p* = 0.342 (**D**) shows CD19^+^ cells, *significantly different from the corresponding control, *p* = 0.028, as measured in the CNS. In (**E**–**H**) the quantitative analysis of the adult animals of (**E**) CD45^int^ cells was performed, ns *p* = 0.885 (**F**) depicts Ly6G^+^ cells, *significantly different from the corresponding control, *p* = 0.028, (**G**) shows CD3^+^ cells,*significantly different from the corresponding control, *p* = 0.028, (**H**) demonstrates CD19^+^ cells, *significantly different from the corresponding control, *p* = 0.028, as measured in the blood is shown. *n* = 4 in each group. All data are given as means ± S.D.

**Figure 8 cells-09-00066-f008:**
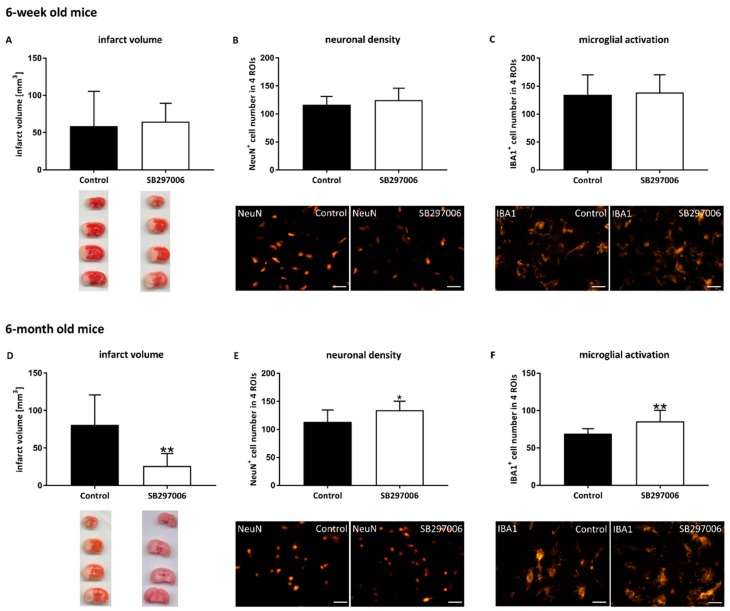
The CCL11 antagonist SB297006 inhibits aggravation of acute ischemic brain injury in adult mice. C57BL6 male mice were subjected to cerebral ischemia for 45 min followed by reperfusion. At the beginning of the reperfusion, mice received intraperitoneal injections of 1 mg/kg bodyweight of the CCL11 antagonist SB297006 or 50 µL of DMSO followed by additional daily injections for three consecutive days. (**A**) Infarct volume measured using TTC staining, ns *p* = 0.780 of adolescent mice. (**B**) Assessment of neuronal density as indicated by Neun^+^ cells, ns *p* = 0.458 of adolescent mice and (**C**) analysis of microglial activation quantifying the number of IBA1^+^ cells, ns *p* = 0.743, of adolescent mice. (**D**) Infarct volume measurement using TTC staining, **significantly different from the corresponding control, *p* = 0.008 of adult mice. (**E**) Assessment of neuronal density as indicated by NeuN^+^ cells, *significantly different from the corresponding control, *p* = 0.029 of adult mice and (**F**) analysis of microglial activation quantifying the number of IBA1^+^ cells, **significantly different from the corresponding control, *p* = 0.009 of adult mice, was done in four ROIs with representative photographs displayed below each diagram. *n* = 9 in the adolescent SB297006 group and *n* = 10 in the adolescent control group, *n* = 12 in the adult SB297006 group and *n* = 8 in the adult control group. All data are given as means ± S.D., scale bars 10 µm.

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
