# Peer review of "CCL11 Differentially Affects Post-Stroke Brain Injury and Neuroregeneration in Mice Depending on Age"

_cells, 2019, doi:10.3390/cells9010066_

Round 1

Reviewer 1 Report

Translational stroke research is a history of disappointments. Hundreds of experimental treatments showing promising results in animal experiments later failed to have a meaningful clinical impact. It is believed that this dilemma at least partially relates to the use of highly standardized stroke animal modes, often focusing on young and healthy animals. In contrast, the typical stroke patient is old.

In their excellent paper, Lieschke et al. go an extra mile with surprising results. They applied CCL11 in stroke. CCL11 is a pro-inflammatory cytokine, for which mixed (positive and negative) results were reported for other CNS conditions. The team did so in young and aged animals, showing a very modest (overall) beneficial effect in young animals but a negative one in old subjects.

Although the results are disappointing, they are of utmost translational importance and relevance to the field. Next to its novelty, a major strength of this well-written paper is that it reports methods and experiments in much clarity and detail, and also provides comprehensive introduction and discussion. The experimental approach is rather conventional, but very robust and well-designed. I think that this is the kind of research we need to avoid translational failure in the future. The paper also contributes to a better understanding of the role of CCL11 in cerebrovascular diseases, although a few (but important) aspects await clarification (see below).

My suggestions for improvements are as follows:

Please report whether the study was randomized and/or featured allocation concealment. Please also report criteria for CBF drop in Laser Doppler flowmetry during MCAO (e.g., CBF -70% in ROI).

Some infarct volumes were calculated in acute stages. It is not clear entirely how the infarct volume was corrected for cerebral edema which is often present in severe strokes at subacute stages. Please clarify (I hope I did not miss this).

Anti-IBA-1 not only stains microglia, but also (peripheral) macrophages, that can be expected at all the measured time points, although to different amounts. Currently, there is no good way to work around this problem in histology, so this is not an error. It must be discussed and the role of macrophages should be considered/discussed.

Please report sample sizes in each figure legend to increase readability.

Sometimes, it feels that missing statistical significance is due to a lack of statistical power. For instance, see Fig. 1A. As mean and SD are presented, a missing statistical significance at such a huge mean difference is likely a power problem. Please generally report p-values. Only p>0.4 makes rather sure that there is no “real world” difference.

Authors could try an area under the curve (AUC) approach over the entire surveillance time for data presented in Fig. 2 (in particular panels C and D). I think there are very good chances for detecting a statistically significant difference albeit coming at the expense of losing the ability to say when exactly this occurred.

Although the paper adds some understanding of CCL11 modes of action, an interesting aspect is missing. Given the systemic administration of the cytokine, it is not unlikely that the peripheral immune system might have taken a “proinflammatory bias”, leading to aggravated outcome. In case such data is available, it would be very interesting to see what peripheral immune cell/cell activation changes are caused by CCL11 administration in naïve young versus aged mice.

Although the paper is very well written, there are a minor number of typos:

“…points [20].Hence…” (line 129, missing blank space)

“…at the acute stage of the brain.” (line 196; shall it read “brain injury”?)

Reference number 21 contains some strange symbols in the name of Dr. Bähr and the word “impairment”. Might be an auto-formatting problem, but please check all references and correct accordingly.

Please double-check the manuscript for additional errors prior to resubmission.

Authors may wish to consider combining some figures as 11 individual figures are a little bit extensive. Some results might also be placed in supplements.

Author contributions are missing. Instead, kind of a placeholder text is presented. Please add the respective information for each author.

I sign my reviews, J. Boltze

Author Response

Point 1: Please report whether the study was randomized and/or featured allocation concealment.

Response 1: Animals were randomly allocated to either treatment paradigms. The surgeon was blinded to the treatment. Besides, the experimenters were blinded to the treatment paradigm as well. This is now mentioned in the revised M&M part.

Point 2: Please also report criteria for CBF drop in Laser Doppler flowmetry during MCAO (e.g., CBF -70% in ROI).

Response 2: Only animals with a dropdown of LDF signal to ≤20% after insertion of the filament were included in this study. Mice reached original CBF values after removal of the filament, i.e., during reperfusion. As expected, there was no difference with regard to the CBF between the various treatment groups, since treatment paradigms were always initiated at the beginning of the reperfusion.

Point 3: Some infarct volumes were calculated in acute stages. It is not clear entirely how the infarct volume was corrected for cerebral edema which is often present in severe strokes at subacute stages. Please clarify (I hope I did not miss this).

Response 3: Infarct volumes were corrected for cerebral edema. The infarct area in each brain slice was calculated by subtracting the area of nonaffected tissue of the ipsilateral hemisphere from the total area of the contralateral hemisphere. This is now mentioned in the manuscript.

Point 4: Anti-IBA-1 not only stains microglia, but also (peripheral) macrophages, that can be expected at all the measured time points, although to different amounts. Currently, there is no good way to work around this problem in histology, so this is not an error. It must be discussed, and the role of macrophages should be considered/discussed.

Response 4: Thank you for this valuable comment, which we have now addressed in the latest version of the paper. Indeed, we did not focus on the role of microglia because of the restriction in histology. This is an important point. We added this in the discussion as “Of note, IBA1 does not only stain resident microglia but also all myeloid-lineage cells including CNS border-associated macrophages as well as blood-borne monocytes/macrophages. Consequently, we could not completely distinguish between these different cell types [1]. This restriction in histology impedes the assessment of microglia-driven inflammation.”

Point 5: Please report sample sizes in each figure legend to increase readability. Sometimes, it feels that missing statistical significance is due to a lack of statistical power. For instance, see Fig. 1A. As mean and SD are presented, a missing statistical significance at such a huge mean difference is likely a power problem. Please generally report p-values. Only p>0.4 makes rather sure that there is no “real world” difference.

Response 5: Sample sizes and p-values are now reported in each figure legend.

Point 6: Authors could try an area under the curve (AUC) approach over the entire surveillance time for data presented in Fig. 2 (in particular panels C and D). I think there are very good chances for detecting a statistically significant difference albeit coming at the expense of losing the ability to say when exactly this occurred.

Response 6: Again, thank you for this advice. We are, unfortunately, not sure which figure the reviewer is pointing at; the (old) figure 2 did not have a “C” or “D”. We guess that the reviewer means figure 3 (behavioral tests). We changed the statistical design (referring to another reviewer) and can now reach a statistical significance. We added the statistical analysis in the materials and methods part and reported p-values in each figure legend.

Point 7: Although the paper adds some understanding of CCL11 modes of action, an interesting aspect is missing. Given the systemic administration of the cytokine, it is not unlikely that the peripheral immune system might have taken a “proinflammatory bias”, leading to aggravated outcome. In case such data is available, it would be very interesting to see what peripheral immune cell/cell activation changes are caused by CCL11 administration in naïve young versus aged mice.

Response 7: In the context of stroke or in terms of ectopic CCL11 administration into naïve rodents, there is hardly any data available showing a proinflammatory bias resulting in an increased brain injury. However, there is a large amount of data demonstrating CCL11 affecting immune cells. Thus, we think that this is a phenomenon, which cannot be excluded. It would be very interesting to investigate this aspect in further studies. The latter was beyond the scope of the present work. As such, we have addressed the issue in the revised discussion of the present paper. Besides, our study was limited by investigating the peripheral immune response only on day 7 and not during the acute stage of the disease.

Minor Points: Typos, Authors may wish to consider combining some figures as 11 individual figures are a little bit extensive. Some results might also be placed in supplements.

Author contributions are missing. Instead, kind of a placeholder text is presented. Please add the respective information for each author.

Response: Typos were corrected. Figures 1 and 2 and also figures 11 and 12 are now merged. Figures 9 and 10 have been placed in the supplementary file for better presentation and to shorten the overall length. Respective information for each author has been included.

Reviewer 2 Report

This comprehensive manuscript by Lieschke et al shows that CCL11 administered exogenously can differentially affect the ischemic mouse brain depending on the age of the mouse. In adolescent mice (6 weeks of age), CCL11 had no effect on the level of injury or neurological deficit following MCAO. However, in adult mice (6 months of age), CCL11 increased infarct volume and worsened neurological deficit. They also show in subsequent experiments at different timepoints that CCL11 could have varying effects on neurogenesis and gliogenesis in the brain as well as inflammatory cell number in the brain and blood. In addition, they also provided information on how CCL11 could affect both autophagic and apoptotic pathways. Lastly, they used a CCR3 antagonist to show a reduction in injury in adult mice, opposite to the effects of CCL11. This manuscript is well written and is a significant body of work. It provides some interesting findings that could help further delineate the roles of CCL11 in the post-ischemic brain.

Major comments:

Ischemic stroke is a condition that occurs in a predominantly aged population, but this study utilizes adolescent and adult mice, not aged mice. There does not appear to be a rationale in the introduction as to why the authors were comparing adolescent and adult mice without including an aged mice group. This rationale should be added. In addition, in the discussion the authors should present a hypothesis for what would occur with CCL11 administration in aged mice post-MCAO given this would be more relevant to human stroke. There are a number of details missing from the methods section to ensure these experiments were conducted with sufficient quality. Additional methodological details that are required are: Line 104, “heating plate”, at what temperature were the mice maintained at and was this regulated? Line 109, it was mentioned that Laser Doppler was used but there are no data presented. Was there a % drop in blood flow in order for mice to be included in the study? What about reperfusion, did you see an increase in CBF when the filament was pulled out and was this different between the CCL11-treated group and control in mice of either age group? Line 116, TTC staining, what % of TTC was used and how long were the sections incubated for and at what temperature? Line 118, were the TTC-stained slices then used for immunohistochemistry? If so, would the process of TTC incubation lead to changes in antigenicity and therefore affect your immunohistochemistry results, particularly since the TTC staining needs to be conducted on fresh tissue and so there would be delay in tissue fixation. What thickness sections were used for the immunohistochemistry (lines 118-9 and 167)? Lines 122-4 and lines 176-7, there are very limited details on the quantitative analysis of the immunohistochemistry. More details needed to be included stating how exactly data were quantified, how was a positive signal detected and analysed (e.g. segmentation, thresholding, cell counting etc). Line 150, there are no details in the Western blotting methods about how the blot was processed, imaged and quantified. These details need to be included. Is it known that both CCL11 and SB297006 can cross the blood-brain barrier? Is this why they were administered intraperitoneally? Figures 1A and 2A, please report the exact p-values of these two graphs. They both have the same profile (CCL11 increasing infarct volume) but one is significant and the other is not. It would be informative to provide the reader with the p-values so they can judge the significance of this. Figure 3C-D, what are the ANOVA results here because while there was only the 3 day timepoint showing significance there may be an interaction between the effects of time and group on the behavioural outcome. It will be important to present these data. Throughout the results, there are a number of comparisons between control and CCL11 treatment e.g. neuronal number, microglia number, neurogenesis, gliogenesis etc. However, what is missing is a sham group that shows what the normal levels of these parameters are, how they are affected by MCAO and then further changed by CCL11. I am not suggesting for the authora to perform these sham experiments, but to provide a limitation in the discussion pointing out the absence of shams and that the effects of MCAO on these parameters compared to sham brains were not investigated. Figure S1 in the supplementary file shows the gating strategy for the flow cytometry experiments. However, an example scatter plot should be shown at each stage to show how the gating strategy was applied. The SB297006 experiments to show the opposite effect compared to CCL11 in adult mice are valuable. However, there is no explanation as to why the injuries in the SB297006 experiment were conducted at 3 days compared to CCL11 which were conducted at 1 day and 28 days. Therefore, it is difficult to compare as the pathophysiology of ischemic injury is so time dependent. Also, it would have been more informative if SB297006 was administered in combination with CCL11 to counteract its effects and show a true reversal of the effect. This is a limitation that needs to be noted. There is no discussion or explanation as to why there may be a difference between the adolescent brain and the adult brain in terms of response to CCL11. What is the actual reason for the differential response? I notice that the absolute infarct volumes are slightly higher in both groups in the adolescent mice compared to the adult mice. Could this be a potential reason? Is there an age-dependent change in CCR3 expression which CCL11 is a ligand for and therefore alter the extent of its effects. Some added discussion regarding this would be beneficial.

Grammatical comments:

Line 51, “neurorestaurative” should be “neurorestorative” Line 85, “analysists” should be “analysts”

Author Response

Point 1: Ischemic stroke is a condition that occurs in a predominantly aged population, but this study utilizes adolescent and adult mice, not aged mice. There does not appear to be a rationale in the introduction as to why the authors were comparing adolescent and adult mice without including an aged mice group. This rationale should be added.

Response 1: Thank you for this valuable comment, which we have now addressed in the latest version of discussion. Indeed, a comparison between elderly mice and human stroke might appear more appropriate in order to reflect the actual clinical situation. This aspect is a shortcoming of our study. In light of lacking studies using CCL11 in adult stroke mice, our study was designed as a proof-of-concept work. Besides, CCL11 is a pro-inflammatory cytokine and possibly harmful effects in elderly mice cannot be excluded, i.e., high mortality rates in elderly stroke mice receiving CCL11 might occur. This aspect is now also mentioned in the discussion section of the paper. Finally, high levels of endogenous CCL11 found in elderly non-ischemic animals are found to be associated with neurodegeneration [2], an aspect we wanted to avoid as well.

Point 2: In addition, in the discussion the authors should present a hypothesis for what would occur with CCL11 administration in aged mice post-MCAO given this would be more relevant to human stroke.

Response 2: The reviewer is right again. As already discussed for point 1, we have discussed both aspects now in the revised discussion.

Point 3: Line 104, “heating plate”, at what temperature were the mice maintained at and was this regulated?

Response 3: During the surgery, the animals were put on a feedback-controlled heating plate using a rectal thermometer to ensure a constant body temperature of 37 °C. After the surgery, the animals were kept in a cage placed on a heating plate (37 °C) over 24 h to maintain temperature of animals during the critical condition of the animals. We now added this in the material and methods part. The information is now given in the text.

Point 4: Line 109, it was mentioned that Laser Doppler was used but there are no data presented. Was there a % drop in blood flow in order for mice to be included in the study? What about reperfusion, did you see an increase in CBF when the filament was pulled out and was this different between the CCL11-treated group and control in mice of either age group?

Response 4: Only animals with a dropdown of LDF signal to ≤20% were included in this study. The CBF after the reperfusion approximated the initial flow. There was no difference between the treatment groups in the CBF as the first doses of the treatment were always applied at the beginning of the reperfusion. We now added this in the materials and methods part.

Point 5: Line 116, TTC staining, what % of TTC was used and how long were the sections incubated for and at what temperature? Line 118, were the TTC-stained slices then used for immunohistochemistry? If so, would the process of TTC incubation lead to changes in antigenicity and therefore affect your immunohistochemistry results, particularly since the TTC staining needs to be conducted on fresh tissue and so there would be delay in tissue fixation. What thickness sections were used for the immunohistochemistry (lines 118-9 and 167)? Lines 122-4 and lines 176-7, there are very limited details on the quantitative analysis of the immunohistochemistry. More details needed to be included stating how exactly data were quantified, how was a positive signal detected and analysed (e.g. segmentation, thresholding, cell counting etc).

Response 5: We apologize for keeping short the description. These methods are commonly used and well established by our research group [3,4]. The information is now fully given in the manuscript. Basically, the brains were removed 24 h after reperfusion and 2 mm coronal sections were dissected. Thereafter, the slices were immediately immersed in 2 % TTC in PBS at 37 °C for 10 min for vital staining and photographed. We generated new animals for immunohistochemistry. Cryostat sections were used with a thickness of 14 µm. Quantitative analyses for immunohistochemical stainings were performed by defining regions of interest (ROIs) within the SVZ and the basal ganglia. The stereotactic coordinates for quantitative analysis in the SVZ were 0.14 mm anterior, 2–3 mm ventral and 1–1.25 mm lateral from bregma. Cell count analysis within the basal ganglia was done at 0.14 mm anterior, 2.5–3.25 mm ventral and 1.5–2.25 mm lateral from bregma. The number of proliferating cells (BrdU+ cells) was investigated within the BG and the SVZ by manual cell counting.

Point 6: Line 150, there are no details in the Western blotting methods about how the blot was processed, imaged and quantified. These details need to be included.

Response 6: Again, we apologize for being so short describing our techniques. All the information necessary to copy our experiments in this respect is now fully given in the appropriate section of the M&M part.

Point 7: Is it known that both CCL11 and SB297006 can cross the blood-brain barrier? Is this why they were administered intraperitoneally?

Response 7: Following previous protocols, we injected CCL11 intraperitoneally which is known to pass the blood-brain barrier as has been reported afore [5][6]. This is also mentioned in the introduction. There is no data available whether or not SB297006 can pass the blood-brain barrier. Moreover, sufficient in vivo data regarding SB297006 in general does not exist. We injected the antagonist intraperitoneally reflecting the CCL11 protocol. Since the antagonist reversed the effects of CCL11, we assumed that SB297006 can indeed pass the blood-brain barrier, especially taking into that the inhibitor was given at times when the blood-brain barrier should be open due to the stroke itself. Nevertheless, we cannot completely exclude any systemic effects, either. We have therefore discussed this aspect in the revised version of the discussion.

Point 8: Figures 1A and 2A, please report the exact p-values of these two graphs. They both have the same profile (CCL11 increasing infarct volume) but one is significant and the other is not. It would be informative to provide the reader with the p-values so they can judge the significance of this. Figure 3C-D, what are the ANOVA results here because while there was only the 3 day timepoint showing significance there may be an interaction between the effects of time and group on the behavioural outcome. It will be important to present these data.

Response 8: This is an extremely good point, which in fact helped us to remove a mistake in the original version of the paper. The p-values are now reported in every figure legend. We carefully repeated our statistical tests, realizing that we have used an incorrect statistical test. We have now added a 2-way-ANOVA analyses in the materials and methods part and reported our new significant results. Again, we are very glad and thankful that the reviewer pointed out this important issue.

Point 9: Throughout the results, there are a number of comparisons between control and CCL11 treatment e.g. neuronal number, microglia number, neurogenesis, gliogenesis etc. However, what is missing is a sham group that shows what the normal levels of these parameters are, how they are affected by MCAO and then further changed by CCL11. I am not suggesting for the authora to perform these sham experiments, but to provide a limitation in the discussion pointing out the absence of shams and that the effects of MCAO on these parameters compared to sham brains were not investigated.

Response 9: We thank the reviewer for sharing his ideas on this matter, which is an interesting aspect. The present study was designed for the comparison between CCL11-treated stroke mice and stroke control mice with regard to a possible therapeutic approach in the future. As such, the premorbid status of the animals was of less concern. Effects of MCAO on chronic brain injury and neuroregeneration compared to sham mice were therefore not investigated. With regard to the existing literature, effects of CCL11 on non-stroke animals have already been published. En detail, CCL11 levels in both the plasma and the CSF significantly increase over time during aging. Ectopic application of CCL11 in young non-stroke mice (2-3 months old) impairs neurogenesis and hippocampal function in these mice. This issue is discussed and cited in the introduction (references 11 and 12 in the manuscript). Along this line, previous data from our own group has shown that C57BL6 sham mice used for our studies are capable of successfully performing our behavioral tests before induction of stroke [4]. We hope that this answer will satisfy the reviewer, although we are well aware of the fact that the ultimate proof on this matter can only yield additional experiments on sham mice. The latter, however, was not requested by the reviewer.

Point 10: Figure S1 in the supplementary file shows the gating strategy for the flow cytometry experiments. However, an example scatter plot should be shown at each stage to show how the gating strategy was applied.

Response 10: We have now added scatter plots to Figure S1 in the supplementary file.

Point 11: The SB297006 experiments to show the opposite effect compared to CCL11 in adult mice are valuable. However, there is no explanation as to why the injuries in the SB297006 experiment were conducted at 3 days compared to CCL11 which were conducted at 1 day and 28 days. Therefore, it is difficult to compare as the pathophysiology of ischemic injury is so time dependent.

Response 11: In this point, the reviewer is completely right. Since we always compare data from the same time point, i.e., treated mice from day 1 with control mice from day 1, this aspect might be neglected. As a matter of fact, the reason for choosing the 3-day survival paradigm was because of no existing in vivo data of SB297006. The latter has so far only been used in in vitro experiments [7] but not in CNS in vivo experiments. We have therefore changed our delivery paradigm to increase the probability of SB297006 biodistribution.

Point 12: Also, it would have been more informative if SB297006 was administered in combination with CCL11 to counteract its effects and show a true reversal of the effect. This is a limitation that needs to be noted.

Response 12: Again, this is a valuably comment which we included to our revised manuscript. We unfortunately did not investigate the combination of both SB297006 and CCL11. This limitation is added to the discussion.

Point 13: There is no discussion or explanation as to why there may be a difference between the adolescent brain and the adult brain in terms of response to CCL11. What is the actual reason for the differential response? I notice that the absolute infarct volumes are slightly higher in both groups in the adolescent mice compared to the adult mice. Could this be a potential reason? Is there an age-dependent change in CCR3 expression which CCL11 is a ligand for and therefore alter the extent of its effects? Some added discussion regarding this would be beneficial.

Response 13: The infarct volumes from the corresponding adolescent and adult groups (Fig. 1A/D) are – as indicated by the reviewer – only slightly different. Indeed, these differences are not significant between each other. A literature search did not yield any reports as to the expression patterns of CCR3 during aging. Yet, this is an interesting idea, which we have mentioned in the revised conclusion section.

Minor Points: Line 51, “neurorestaurative” should be “neurorestorative” Line 85, “analysists” should be “analysts”

Response: Typos were corrected.

Reviewer 3 Report

Great effort by all the respective authors. Excellent illustrations.

I would like to highlight the following points in the study

Introduction especially the section on CCL11 (56-72) looks lengthy and can be cut short for better presentation Excellent figures and illustrations in results but would recommend cutting down on the written text to shorten the overall length Longer term effects in the treatment mice arm  even after the CCL11 delivery ceased post 7 days ?

Author Response

Point 1: Introduction especially the section on CCL11 (56-72) looks lengthy and can be cut short for better presentation

Response 1: We have carefully shortened this part in the revised version of the manuscript.

Point 2: Excellent figures and illustrations in results but would recommend cutting down on the written text to shorten the overall length

Response 2: Figures 1 and 2 and also figures 11 and 12 were merged. Figures 9 and 10 were placed in the supplementary file for better presentation and to shorten the overall length.

Point 3: Longer term effects in the treatment mice arm even after the CCL11 delivery ceased post 7 days?

Response 3: We are not sure, if we get the reviewers point right. We found out that CCL11 impairs neurological recovery during the administration of CCL11 in the adult animals. When CCL11 delivery has been stopped, the former aggravation of brain injury and neurological impairment was reversed during the observation period of 28 days. In this context, we cannot exclude the possibility of CCL11 having long-lasting effects on post-stroke neuroregeneration beyond day 28. However, such effects are very unlikely when taking into account a high cell death rate of new-born cells after induction of stroke. As a matter of fact, the majority of such cells dies within 3 months post-stroke due to secondary cell death [4]. We included this point in the revised version of the manuscript.

Round 2

Reviewer 2 Report

The authors have adequately addressed all my comments and concerns and I am happy with the revisions made.